# Safety and efficacy of esaxerenone in Japanese hypertensive patients with heart failure with reduced ejection fraction: A retrospective study

**Togo Iwahana**[ORCID]*, **Yuichi Saito, Sho Okada, Hirotoshi Kato, Ryohei Ono, Yoshio Kobayashi**

Department of Cardiovascular Medicine, Chiba University Graduate School of Medicine, Chiba, Japan

* adxa2698@chiba-u.jp

**Data Availability Statement:** All relevant data are within the manuscript and its Supporting Information files.

## Abstract

Esaxerenone, a mineralocorticoid receptor blocker (MRB), is a new antihypertensive agent. However, esaxerenone-related data with respect to hypertension with heart failure are limited. We investigated the safety and efficacy of esaxerenone in hypertensive patients with heart failure with reduced ejection fraction (HFrEF). Hypertensive patients with HFrEF treated with esaxerenone were retrospectively analyzed at two timepoints (short-term: 35 ±15 days; mid-term: 167±45 days). Adverse events including hyperkalemia ($K^+$ >5.5 mEq/L), worsening renal function (WRF; estimated glomerular filtration rate (eGFR) reduction by $\geq$20%), and hypotension (systolic blood pressure <90 mmHg) were evaluated. eGFR and $K^+$, serum creatinine, and brain natriuretic peptide (BNP) levels at baseline, short-term, and mid-term assessments were compared. Patients administered esaxerenone as their first MRB (first-MRB cohort) and those who converted from another MRB (conversion cohort) were separately analyzed. There were 50 (56±10 years old, 26% female) patients. At the short-term assessment, hyperkalemia or hypotension was not observed at a dose of 2.0 ±0.9 mg/day. Seven patients (14%) showed WRF. $K^+$ was slightly elevated (4.12±0.41 to 4.25±0.39 mmol/L, $p = 0.07$) and eGFR was significantly reduced (66.9±19.6 mL/min/1.73 m² to 62.4±19.7 mL/min/1.73 m², $p = 0.006$). In the conversion cohort, significant changes in $K^+$ and eGFR from baseline were not observed at the short-term assessment. BNP levels were consistently improved regardless of the cohorts (first-MRB cohort, 310 [110–370] pg/mL to 137 [47–152] pg/mL, $p = 0.001$; conversion cohort, 181 [30–203] pg/mL to 108 [26–146] pg/mL, $p = 0.028$). At the mid-term assessment, there were no significant changes in $K^+$ and eGFR compared with the short-term assessment. In conclusion, esaxerenone was safe for hypertensive patients with HFrEF. Hyperkalemia and hypotension were rarely noted, while eGFR was marginally reduced. Moreover, esaxerenone might be beneficial for HFrEF in terms of BNP level reduction.

**Funding:** This work was supported by JSPS KAKENHI Grant Number JP19K17514 to TI (http://www.jsps.go.jp/english/). The funder had no role in study design, data collection and analysis, decision to publish, or preparation of the manuscript.

**Competing interests:** We declared that YK received lecture fees and research grant from Daiichi Sankyo Co., Ltd. YK does not have any other relation with this funder, including employment, consultancy, patents, products in development, or marketed products. This does not influence our adherence to PLOS ONE policies on sharing data and materials.

## Introduction

Hypertension is a major risk factor for heart failure (HF) [1]. Mineralocorticoid receptor (MR)-associated hypertension, in which MR is excessively stimulated, is detected in a proportion of patients presenting with resistant hypertension [2]. In patients with MR-associated hypertension, MR blockers (MRBs) with first-line antihypertensive agents, such as angiotensin-converting enzyme (ACE) inhibitors, angiotensin receptor blockers (ARBs), calcium channel blockers, and diuretics have been suggested to be effective in controlling blood pressure (BP) [3].

MRBs have also improved the prognosis of HF with reduced ejection fraction (HFrEF) in previous studies, such as RALES [4], EPHESUS [5], and EMPHASIS-HF [6]. Administration of MRBs with ACE inhibitors or ARBs, and β-blockers for patients with HFrEF has been recommended according to the guidelines [7,8].

Recently, the FIDELIO-DKD trial investigating the kidney and cardiovascular outcomes and safety of finerenone, a novel non-steroidal MRB, compared with a placebo in patients with type 2 diabetes and chronic kidney disease with albuminuria was published [9]. Patients in the finerenone group showed a significant decrease of kidney disease progression or cardiovascular death compared to the placebo group, suggesting that MRBs improve both the renal and cardiovascular prognosis in patients with diabetic nephropathy [9].

Although MRBs are essential in the treatment of resistant hypertension and HFrEF, they have several problems. Hyperkalemia is one of the most important adverse events (AEs). In addition, spironolactone results in gynecomastia [4], while eplerenone is contraindicated in diabetic patients with microalbuminuria or proteinuria and in those with moderate or severe renal dysfunction (creatinine clearance <50 mL/min) [10]. Furthermore, hypotension is also an issue in the treatment of HF.

Esaxerenone is a novel, non-steroidal, and selective MRB [11]. It was developed in Japan and was approved as an antihypertensive agent [11]. The ESAX-HTN phase 3 trial [12] has reported that treatment with esaxerenone (2.5 and 5 mg/day) in patients with essential hypertension is non-inferior to that with eplerenone (50 mg/day). Moreover, the incidence of AEs, such as hyperkalemia and worsening of renal failure (WRF), was similar between the two.

Esaxerenone is expected to be beneficial in patients with HF; however, it is only indicated for hypertensive patients in Japan. Data on esaxerenone for HF, especially for HFrEF, are limited. In the present study, we investigated the short- to mid-term safety and efficacy of esaxerenone in patients with hypertension with HFrEF.

## Materials and methods

### Study design

This was a single-center, retrospective observational study conducted in Chiba University Hospital. In order to investigate the safety and efficacy of esaxerenone for hypertensive patients with HFrEF, laboratory and BP data obtained before and after the administration of esaxerenone were retrospectively analyzed. This study protocol was approved by the Ethics Committee of Chiba University in accordance with the 2013 Declaration of Helsinki.

### Patient selection

Hypertensive patients with HFrEF who were newly treated with esaxerenone at the Chiba University Hospital between July 2019 and January 2021 were retrospectively included. Hypertension was defined as a systolic BP (SBP) >140 mmHg and/or diastolic BP (DBP) >90 mmHg or those who were already prescribed with antihypertensive agents due to a diagnosis of

hypertension. The criteria for HFrEF were as follows: (i) left ventricular ejection fraction (LVEF) <40%, and (ii) New York Heart Association (NYHA) class II, III, or IV. Patients with baseline serum potassium ($K^+$) levels >5 mEq/L or those lost to follow-up were excluded. All participants were given the option to opt out of the study through a notice on the website.

## Study protocol

Baseline characteristics including age, sex, body mass index (BMI), NYHA class, etiology of HF, BP, heart rate (HR), comorbidities, current antihypertensive agents, laboratory data including serum creatinine (Cre), $K^+$, and brain natriuretic peptide (BNP) levels, and echocardiographic data were collected from medical records. The estimated glomerular filtration rate (eGFR) was calculated using the Japanese GFR equation: $eGFR = 194 \times Cre^{1.094} \times Age^{0.287} \times 0.739$ (if female) [13].

Esaxerenone was usually started at a dose of 2.5 mg daily. However, in patients with renal dysfunction (eGFR <60 mL/min/1.73 m$^2$) or low cardiac function, the initial dose was 1.25 mg daily according to their physicians' discretion. The dose was titrated by monitoring BP and blood examination to a maximum dose of 5 mg daily.

In the short-term assessment, BP and laboratory data obtained at the first evaluation two weeks after the initiation of esaxerenone were compared with baseline data. Mid-term assessment was performed at 6±2 months after the initiation of esaxerenone in patients whose data were available. The primary endpoint was the incidence of AEs, such as hyperkalemia, WRF, hypotension, and cardiovascular events. The secondary endpoint was changes in blood examination findings, such as changes in $K^+$, Cre, eGFR, and BNP levels. For patients whose data were available, the changes in BP were also analyzed (Fig 1).

As a sub-analysis, patients who were converted from another MRB to esaxerenone (conversion cohort) and patients who took esaxerenone as their first MRB (first MRB cohort) were separately analyzed (Fig 1).

## Definition of AEs

Hyperkalemia was defined as $K^+$ >5.5 mEq/L, WRF as ≥20% in eGFR decrease, and hypotension as SBP <90 mmHg at clinic and/or home BP measurement. Cardiovascular event was defined as an all-cause death and hospitalization due to worsening HF.

## Statistical analysis

Continuous variables were expressed as mean ± standard deviation (SD) and were compared using the paired t-test. BNP was expressed as median [IQR] and compared using the Wilcoxon signed-rank test. Categorical variables were expressed as numbers and percentages and were compared using the chi-squared test. Statistical significance was set at $p < 0.05$. Processing and statistical analyses were performed using JMP$^®$ Pro.14.2.0 (SAS Institute, Cary, NC).

## Results

### Baseline characteristics

A total of 93 patients with hypertension and HF with newly introduced esaxerenone were identified. The following patients were excluded: 35 patients with LVEF ≥40%, 5 patients lacking follow-up laboratory data, and 3 patients who discontinued esaxerenone within 2 weeks. Thus, 50 patients were included in this study. The reasons for the three patients' esaxerenone discontinuation within 2 weeks were $K^+$ elevation, eGFR reduction (not reaching the AE

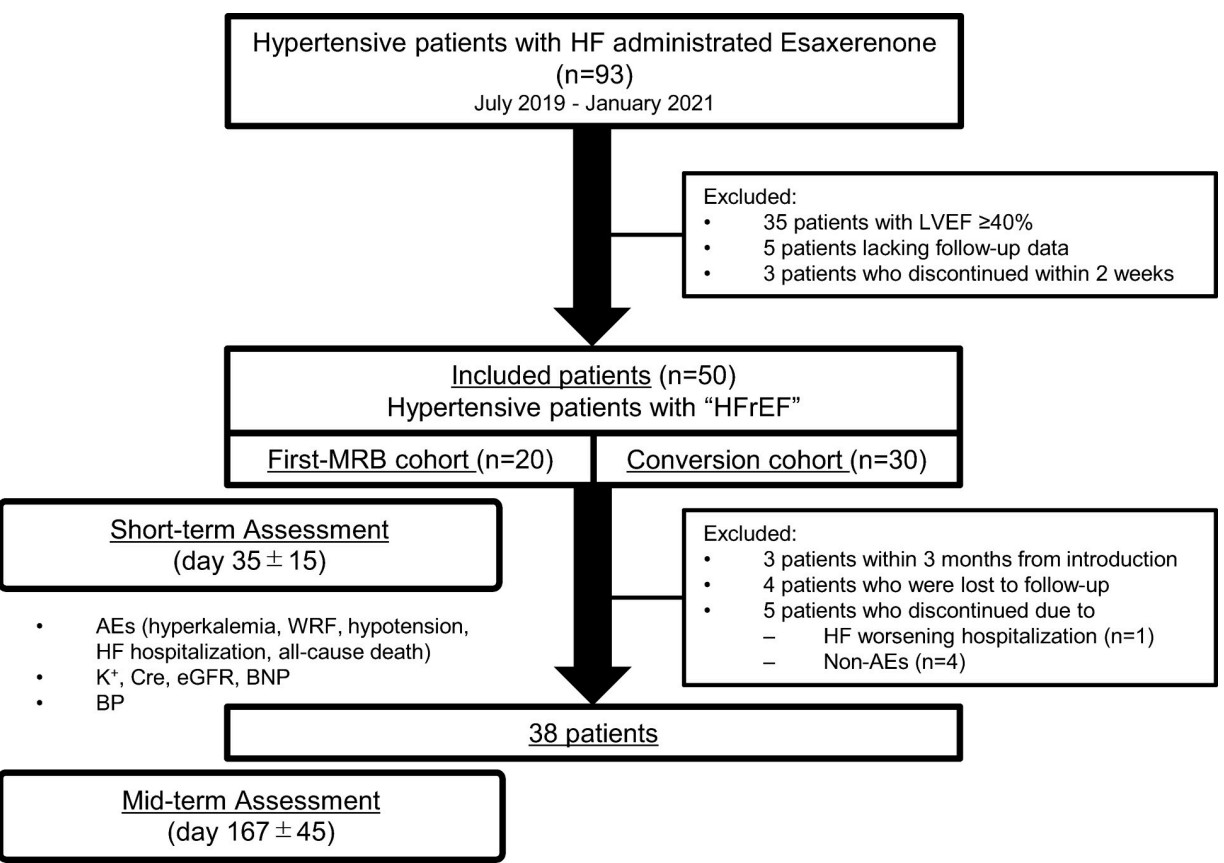

**Fig 1. Study protocol.** The first-MRB cohort was defined as patients who have not been treated with other MRBs. The conversion cohort was defined as patients who were converted from another MRB. HFrEF, heart failure with reduced ejection fraction; LVEF, left ventricular ejection fraction; MRB, mineralocorticoid receptor blocker; WRF, worsening of renal function; $K^+$, serum potassium; Cre, serum creatinine; eGFR, estimated glomerular filtration rate; BNP, brain natriuretic peptide; BP, blood pressure; AE, adverse event.

criterion), and planned cardiac surgery, respectively. A total of 30 patients (60%) were converted from another MRB. Baseline characteristics are presented in Table 1.

## Short-term assessment

In the short-term assessment at 35±15 days after initiation of esaxerenone, the average dose of esaxerenone was 2.0±0.9 mg/day. The number of antihypertensive agents taken by patients significantly increased; however, the usage rates of each antihypertensive agent except for MRBs were similar (Table 2). The dose of β-blockers (carvedilol equivalent dose) was slightly reduced and that of loop diuretics (furosemide equivalent dose) was similar (Table 2).

No patients were deemed hyperkalemic. The mean $K^+$ elevation was 0.13±0.50 mEq/L, which was not significant (Fig 2A). WRF was observed in 7 patients (14%), and the mean changes in Cre and eGFR were 0.06±0.17 mg/dL ($p = 0.01$) and -4.5±10.9 mL/min/1.73 $m^2$ ($p<0.01$), respectively (Fig 2B and 2C). BNP levels significantly decreased from 232 [45–354] pg/mL to 120 [37–149] pg/mL ($p<0.01$) (Fig 2D). Clinic BP data of 37 patients (74%) were available. Home BP records were also checked to evaluate the presence of hypotension. SBP was slightly decreased with marginal significance (128±22 mmHg to 121±16 mmHg, $p = 0.06$), whereas DBP was similar (78±15 mmHg to 75±15 mmHg, $p = 0.78$). None of the patients presented with hypotension.

**Table 1. Baseline patient characteristics.**

| | |
|---|---:|
| Age, years old | 55.9 ± 15.0 |
| Male, n (%) | 37 (74%) |
| Body weight, kg | 75.0 ± 19.4 |
| BMI, kg/m$^2$ | 26.3 ± 6.2 |
| NYHA 3/4, n (%) | 14 (28%) |
| ICM, n (%) | 7 (14%) |
| Non-ICM, n (%) | 43 (86%) |
| Comorbidities | |
| AF, n (%) | 14 (28%) |
| DM, n (%) | 12 (24%) |
| Dyslipidemia, n (%) | 24 (48%) |
| CKD, n (%) | 15 (30%) |
| Device Therapies | |
| ICD, n (%) | 4 (8%) |
| CRT/CRT-D, n (%) | 7 (14%) |
| Antihypertensive Agents | |
| ACE inhibitors/ARBs, n (%) | 43 (86%) |
| CCBs, n (%) | 9 (18%) |
| Diuretics, n (%) | 40 (80%) |
| MRBs, n (%) | 30 (60%) |
| Spironolactone, n (%) | 21 (42%) |
| Eplerenone, n (%) | 9 (18%) |
| β-blockers, n (%) | 44 (88%) |
| α-blockers, n (%) | 1 (2%) |
| Number of antihypertension agents, n | 3.4 ± 0.9 |
| Systolic blood pressure, mmHg | 128 ± 22 |
| Diastolic blood pressure, mmHg | 78 ± 15 |
| Heart rate, bpm | 79 ± 15 |
| Laboratory | |
| Cre, mg/dL | 0.95 ± 0.34 |
| eGFR, mL/min/1.73 m$^2$ | 66.9 ± 19.6 |
| K, mEq/L | 4.1 ± 0.4 |
| BNP, pg/mL | 167 [63–375] |
| Echocardiography | |
| LVEF, % | 31 ± 6 |
| LVDD, mm | 62 ± 8 |
| LVDS, mm | 53 ± 8 |
| IVST, mm | 9 ± 2 |
| LVPW, mm | 10 ± 2 |
| LAVI, mL/m$^2$ | 46.5 ± 22.1 |
| E/A | 1.19 ± 1.00 |
| E/e' | 13.0 ± 7.0 |

Data are presented as the mean ± SD unless specified. BNP was expressed as the median [IQR].

Abbreviations: BMI, body mass index; NYHA, New York Heart Association functional class; AF, atrial fibrillation; DM, diabetes mellitus; CKD, chronic kidney disease; ICD, implantable cardiac defibrillator; CRT, cardiac resynchronization therapy; CRT-D, cardiac resynchronization therapy defibrillator; ACE, angiotensin converting enzyme; ARB angiotensin II receptor blocker; CCB, calcium channel blocker; MRB, mineralocorticoid receptor blocker; Cre, serum creatinine; eGFR, estimated glomerular filtration rate; K, serum potassium; BNP, plasma brain natriuretic peptide; LVEF, left ventricular ejection fraction; LVDD, left ventricular end-diastolic diameter; LVDS, left ventricular end-systolic diameter; IVST, interventricular septal thickness; LVPW, left ventricular posterior wall; LAVI, left atrial volume index.

**Table 2. Comparison of medication between baseline and the short-term assessment.**

|  | Baseline | Short-term assessment | p value |
|---|---|---|---|
| ACE inhibitors or ARBs | 43 (86%) | 46 (92%) | 0.34 |
| ACE inhibitors, n (%) | 35 (70%) | 38 (76%) | 0.50 |
| ARBs, n (%) | 8 (16%) | 8 (16%) | 1.00 |
| β-blockers, n (%) | 44 (88%) | 48 (96%) | 0.14 |
| carvedilol equivalent dose, mg | 12.2 ± 7.7 | 12.0 ± 7.5 | 0.03 |
| MRBs, n (%) | 30 (60%) | 50 (100%) | <0.01 |
| spironolactone, n (%) | 21 (42%) | 0 (0%) |  |
| eplerenone, n (%) | 9 (18%) | 0 (0%) |  |
| esaxerenone, n (%) | 0 (0%) | 50 (100%) |  |
| Diuretics, n (%) | 40 (80%) | 40 (80%) | 1.00 |
| thiazide, n (%) | 6 (12%) | 5 (10%) | 0.75 |
| loop, n (%) | 36 (72%) | 38 (76%) | 0.65 |
| furosemide equivalent dose, mg | 31.1 ± 17.2 | 31.1 ± 16.9 | 0.79 |
| CCBs, n (%) | 9 (18%) | 9 (18%) | 1.00 |
| α-blockers, n (%) | 1 (2%) | 0 (0%) | 0.31 |
| Number of antihypertensive agents (total) | 3.4 ± 0.9 | 3.9 ± 0.7 | <0.01 |
| Number of antihypertensive agents (except for MRBs) | 2.8 ± 0.7 | 2.9 ± 0.7 | 0.051 |

Data are mean ± SD unless specified. Categorical variables are expressed as numbers and percentages and were compared using the chi-squared test. Continuous variables were compared using the paired t-test.

Abbreviations: ACE, angiotensin converting enzyme; ARB, angiotensin II receptor blocker; CCB, calcium channel blocker; MRB, mineralocorticoid receptor blocker.

### Sub-analysis on the first-MRB and conversion cohorts

In the first-MRB cohort (n = 20) in the short-term assessment, the changes in $K^+$, Cre, and eGFR were significant (Fig 3A–3C). Six patients (30%) presented with WRF; however, none had hyperkalemia. Analysis of BP in the first-MRB cohort (n = 16) showed a significant decrease in SBP (138±24 mmHg to 127±18 mmHg, $p$ = 0.03).

The short-term assessment in the conversion cohort (n = 30) showed that there was no significant change in $K^+$, Cre, or eGFR compared with their baseline values (Fig 3A–3C). In both cohorts, BNP levels were consistently lowered (Fig 3D).

### Mid-term assessment

The mid-term assessment was performed in 38 patients with data obtained at 167±45 days after initiation of esaxerenone. Esaxerenone was titrated to 2.8±1.3 mg/day ($p$<0.01), and the number of antihypertensive agents was similar to that in the short-term assessment (4.0±0.6, $p$ = 0.71). Six patients (16%) presented with WRF, and none had hyperkalemia. Neither $K^+$, Cre, nor eGFR changed compared to the short-term assessment (Fig 4A–4C). BNP was consistently lowered compared to that in the short-term assessment (Fig 4D).

### Discussion

In the present study, we investigated the safety and efficacy of esaxerenone in hypertensive patients with HFrEF. The main findings were as follows: (i) $K^+$ level was slightly elevated with mild eGFR reduction; (ii) significant changes in $K^+$ levels and renal function were not observed in patients converted from another MRB; and (iii) BNP was consistently decreased regardless of previous MRB treatment.

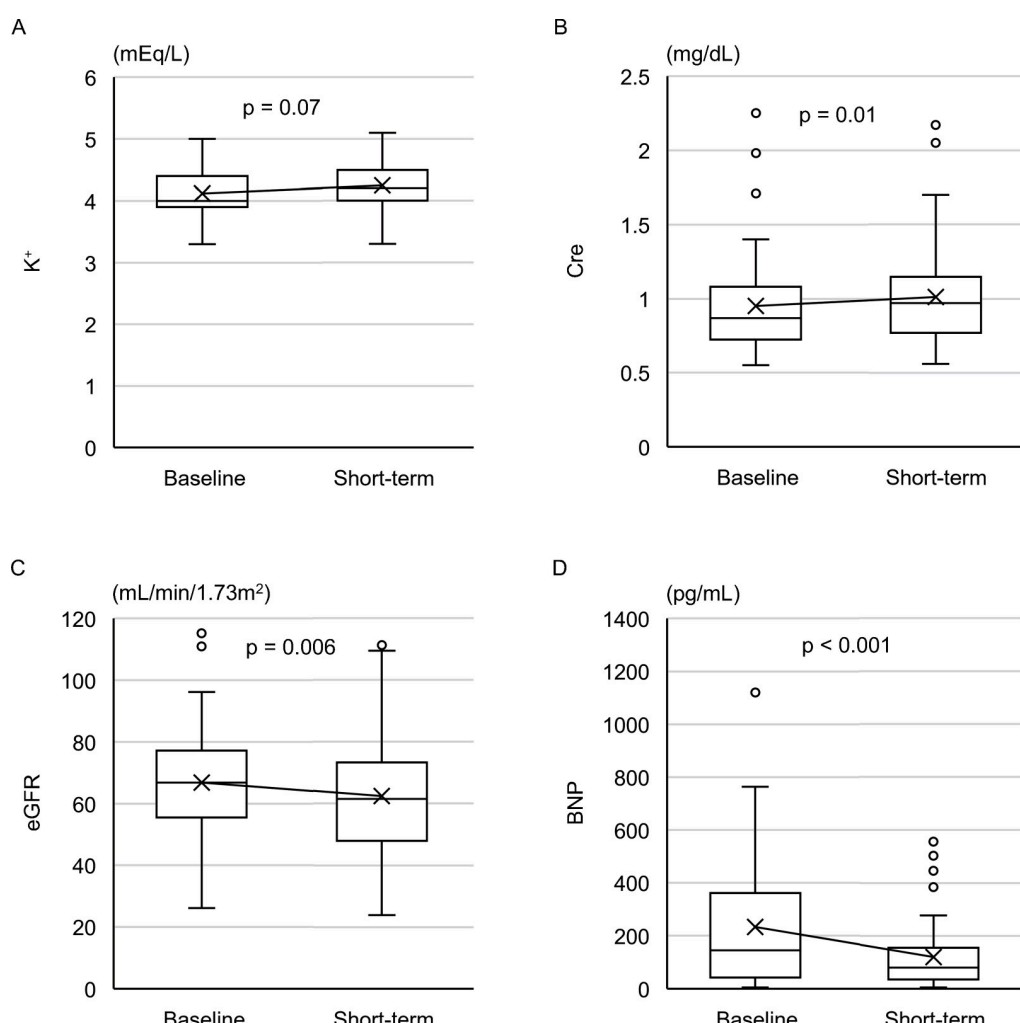

**Fig 2. The short-term assessment for all included patients.** Changes in K⁺ (A), Cre (B), eGFR (C), and BNP (D) between baseline and short-term assessments (all patients included). K+, Cre, eGFR were compared using the paired t-test, and BNP level was compared using the Wilcoxon signed-rank test. K⁺, serum potassium; Cre, serum creatinine; eGFR, estimated glomerular filtration rate; BNP, brain natriuretic peptide.

There are limited data on the usage of esaxerenone in the treatment of HF. Naruke et al. firstly reported detailed changes in BP and treatment-related AEs in hypertensive patients with HF [14]. This retrospective study included patients regardless of LVEF, where the majority (82%) of patients had LVEF ≥40%. Since MRBs other than esaxerenone have been shown to be effective for HFrEF, analyses of esaxerenone focused on HFrEF are required. However, there are concerns of MRB-induced hypotension and peripheral circulatory failure, especially in patients with HFrEF. Therefore, we mainly investigated the safety of esaxerenone as well as efficacy focused in hypertensive patients with HFrEF. To our knowledge, this is the first report regarding the safety and efficacy of esaxerenone in HFrEF patients. Moreover, our study included a number of patients who were converted from another MRB.

## Safety for K⁺ levels and renal function

Hyperkalemia is one of the most important AEs of MRB [15]. EMPHASIS-HF [6], a study investigating the efficacy and safety of eplerenone for symptomatic HFrEF patients, showed

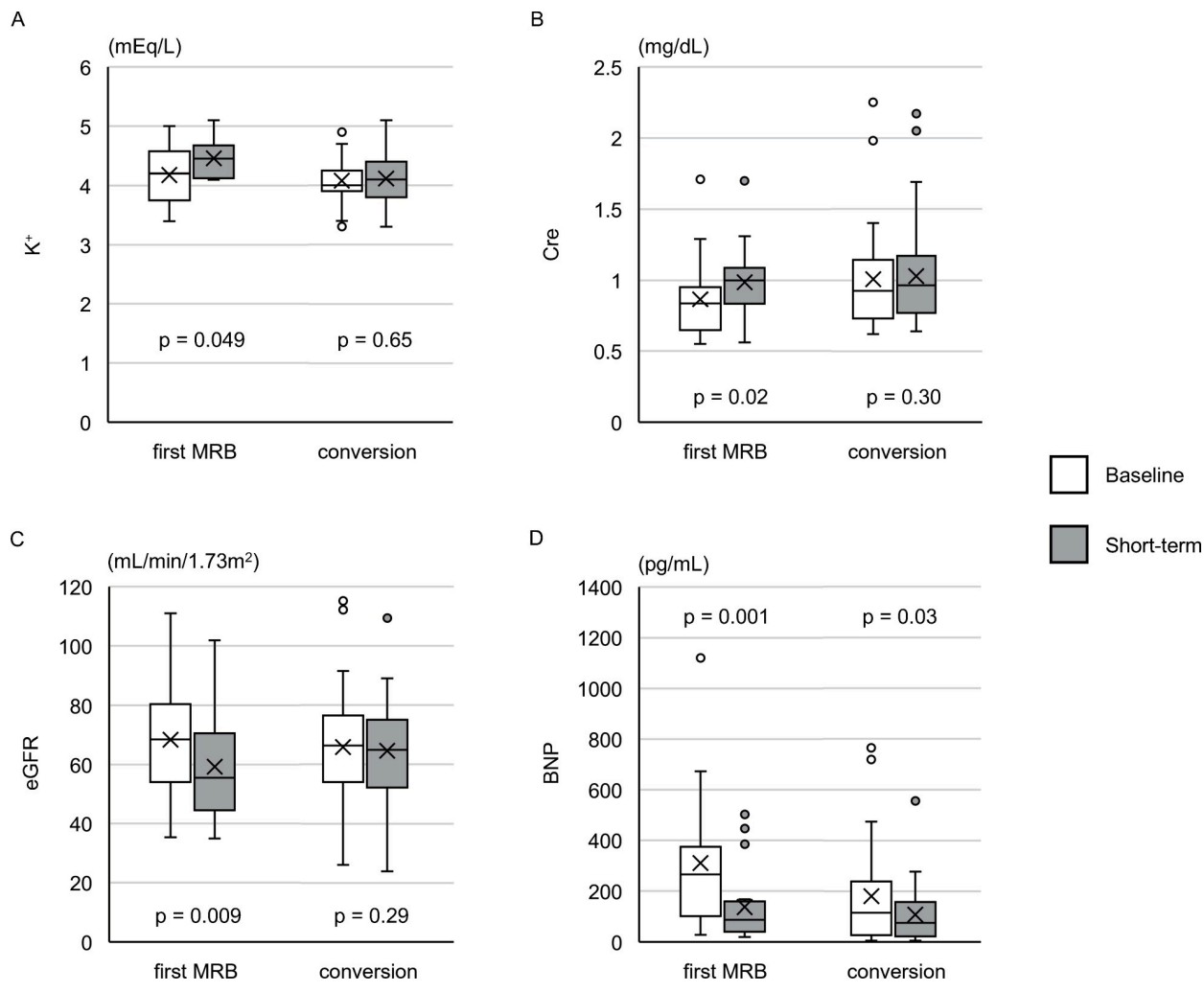

**Fig 3. The short-term assessment for the first-MRB and conversion cohorts.** Changes in K+ (A), Cre (B), eGFR (C), and BNP (D) between baseline and short-term assessments in the first-MRB (n = 20) and conversion (n = 30) cohorts. In each cohort, left bar indicates the baseline, and the right bar indicates the short-term assessment. K+, Cre, eGFR were compared using the paired t-test, and BNP level was compared using the Wilcoxon signed-rank test. K+, serum potassium; Cre, serum creatinine; eGFR, estimated glomerular filtration rate; BNP, brain natriuretic peptide.

that the mean change in K+ level after a month was 0.16±0.51 mEq/L. Hyperkalemia (K+>5.5 mEq/L) occurred in 11.8% [6]. Several phase III clinical trials of esaxerenone, which included patients with essential hypertension, hypertension with renal dysfunction, type 2 diabetes, or primary aldosteronism, but all without HF, showed that hyperkalemia occurred in 1.7% of esaxerenone-treated patients (including unpublished data) [12,16–18]. In the present study, no patient showed hyperkalemia, although K+ was similarly elevated by 0.28±0.59 mEq/L in the first-MRB cohort in the short-term assessment. It has been reported that MRB treatment causes a temporary eGFR decrease by relieving glomerular hyperfiltration [19]. However, MRBs had significant antiproteinuric and antialbuminuric effects and improved cardiorenal prognosis when added to ACEIs or ARBs [20]. In the present study, patients in the first-MRB cohort showed an increase in Cre by 0.12±0.22 mg/dL, which was comparable with the results of the EMPHASIS-HF trial (0.15±0.35 mg/dL) [6]. It should be noted that 30% of patients presented with WRF in the first-MRB cohort. In fact, five of the six patients presenting with WRF started on 2.5 mg of esaxerenone. To avoid a sharp change in renal function, initiation from 1.25 mg seems to be safer.

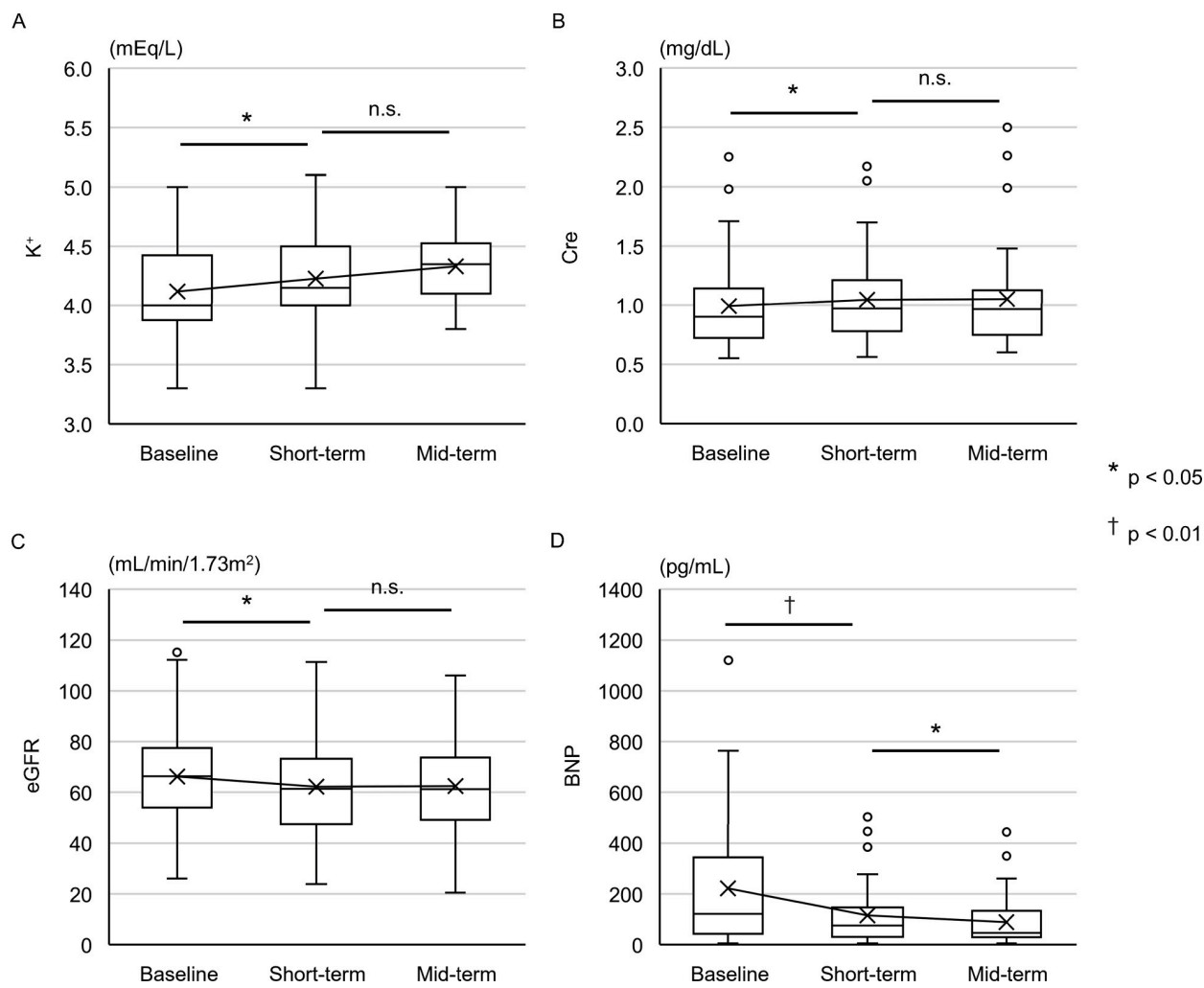

**Fig 4. The mid-term assessment of 38 patients.** Changes in $K^+$ (A), Cre (B), eGFR (C), and BNP (D) between baseline, short-term, and mid-term assessments for 38 patients with mid-term assessment data available. $K^+$, Cre, eGFR were compared using the paired t-test, and BNP level was compared using the Wilcoxon signed-rank test. *: $p < 0.05$, and †: $p < 0.01$. $K^+$, serum potassium; Cre, serum creatinine; eGFR, estimated glomerular filtration rate; BNP, brain natriuretic peptide.

In the conversion cohort, $K^+$, Cre, and eGFR did not significantly change, suggesting that the risks of esaxerenone in altering the $K^+$ levels and renal function are not striking compared with spironolactone and eplerenone.

## Safety for blood pressure

For patients with HFrEF, ACE inhibitors or ARBs, β-blockers, and MRBs are recommended [7,8]. However, adding MRBs could lead to hypotension and circulatory failure. In the present study, no patient presented with hypotension, even if in some patients SBP had been already normalized at baseline with other antihypertensive agents. It was suggested that low-dose esaxerenone rarely decreases BP excessively.

## Efficacy for HF

Regardless of the cohort, BNP consistently decreased over time. According to a review analyzing 16 phase III chronic HF trials, therapy-related changes in natriuretic peptides appeared

modestly correlated with longer-term therapeutic effects on hospitalization for HF, but not with effects on all-cause mortality [21]. Regarding MRBs, the ARTS-HF study reported that 30–40% of HF patients with diabetes mellitus and/or chronic kidney disease treated with eplerenone or finerenone presented with a >30% reduction of NT-proBNP [22]. Esaxerenone in the present study demonstrated comparable BNP reduction to other MRBs, which suggests that esaxerenone is potentially beneficial for HFrEF.

## Study limitations

The present study has several limitations. First, this non-randomized observational study without a control group was performed at a single center with a small number of patients for a limited observation period and a Japanese-specific population. Indeed, esaxerenone is currently available only in Japan. Moreover, we were unable to follow-up the patients long-term, which seems to be one reason why the number of AEs was small. In previous studies regarding MRBs, however, $K^+$ elevation and eGFR decrease were reported to be most conspicuous within the first month [18,23]. We intended to investigate mainly the safety of the drug; thus, our results have a certain significance regarding safety. A large-scale, multinational, and randomized control study is required to resolve the potential bias. Second, clinic BP data were unavailable in a number of patients. In these patients, home BP records were checked to diagnose hypotension. Based on the medical records, none of the patients discontinued esaxerenone due to hypotension or dizziness. The main purpose of this study was to assess the drug's safety, which was accomplished with hypotension as the endpoint. Third, the combination of antihypertensive agents other than MRBs was somewhat variable; however, there was no significant difference in the number of antihypertensive agents other than MRBs. Fourth, a minority of patients underwent treatment for acute decompensated HF with intravenous vasodilators and/or diuretics at baseline. BP and BNP levels may be affected by factors other than esaxerenone.

## Conclusions

The present study suggested that esaxerenone is safe for hypertensive patients with HFrEF, with a low incidence of hyperkalemia and hypotension. Changes in renal function and incidence of WRF were acceptable and similar to those observed in patients treated with spironolactone and eplerenone. Moreover, esaxerenone was found to be beneficial for HF in terms of BNP level reduction.

## Supporting information

**S1 Table. Study data.** Patients' dataset.
(XLSX)

## Acknowledgments

We would like to thank Editage (www.editage.jp) for English language editing.

## Author Contributions

**Conceptualization:** Togo Iwahana, Yuichi Saito, Sho Okada, Yoshio Kobayashi.

**Data curation:** Yuichi Saito, Ryohei Ono.

**Formal analysis:** Togo Iwahana.

**Funding acquisition:** Togo Iwahana, Yoshio Kobayashi.

**Investigation:** Togo Iwahana, Hirotoshi Kato, Ryohei Ono.

**Supervision:** Yuichi Saito, Sho Okada, Yoshio Kobayashi.

**Visualization:** Togo Iwahana.

**Writing – original draft:** Togo Iwahana.

**Writing – review & editing:** Yuichi Saito, Sho Okada, Hirotoshi Kato, Ryohei Ono, Yoshio Kobayashi.

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
