## [Decision Letter · Decision Letter 0]

23 Aug 2021

PONE-D-21-24007

Safety and Efficacy of Esaxerenone in Hypertensive Patients with Heart Failure with Reduced Ejection Fraction

PLOS ONE

Dear Dr. Iwahana,

Thank you for submitting your manuscript to PLOS ONE. After careful consideration, we feel that it has merit but does not fully meet PLOS ONE’s publication criteria as it currently stands. Therefore, we invite you to submit a revised version of the manuscript that addresses the points raised during the review process.

ACADEMIC EDITOR: All issues raised by expert reviewers are required.

We look forward to receiving your revised manuscript.

Kind regards,

Vincenzo Lionetti, M.D., PhD

Academic Editor

PLOS ONE

Journal Requirements:

a) Did participants provide their written or verbal informed consent to participate in this study?

3. Thank you for stating the following in the Competing Interests/Financial Disclosure * (delete as necessary) section: 

" I have read the journal's policy and the authors of this manuscript have the following competing interests: YK received lecture fees and research grant from Daiichi Sankyo Co., Ltd. TI, YS, SO, HK, and RO have no conflicts of interest to declare"

We note that you received funding from a commercial source: Daiichi Sankyo Co., Ltd

Reviewers' comments:

Reviewer's Responses to Questions

**Comments to the Author**

1. Is the manuscript technically sound, and do the data support the conclusions?

Reviewer #1: Partly

Reviewer #2: Yes

2. Has the statistical analysis been performed appropriately and rigorously? 

Reviewer #1: Yes

Reviewer #2: Yes

3. Have the authors made all data underlying the findings in their manuscript fully available?

Reviewer #1: Yes

Reviewer #2: Yes

4. Is the manuscript presented in an intelligible fashion and written in standard English?

Reviewer #1: Yes

Reviewer #2: Yes

5. Review Comments to the Author

Reviewer #1: Comment: Iwahana et al. conducted a retrospective study to investigate the safety and the efficacy of esaxerenone, a novel non-steroidal MRB in subjects with hypertension and heart failure with reduced ejection fraction. After a short term (35 days) and mid term (167 days) follow-up, no serious adverse event (hypotension, hyperkalemia, hospitalization for health failure) was observed. A small number (n=7) of patients showed a worsening of renal function, though superimposable with the one reported in other major trials and of mild clinical relevance. A slight, non-significant, elevation of potassium was also detected. In terms of efficacy, however, a consistent over time BNP reduction was assessed.

Albeit with the limitations implicit in a small retrospective study, the novelty of the drug under observation and the significance of the clinical context make the findings and the paper relevant. However, the authors should amend some flaws, mainly in the discussion section, to make it suitable for publication.

Major

Limitations: the main limits of this work lie in the combination of being a retrospective study with a very small sample size and no control group. Indeed, this substantially hinder the appreciation of an efficacy effect, and this should be wider discussed in the “study limitation”. Also, cardiovascular events were defined as outcome. However it seems, from the text, that no event was reported (except that one in Figure 1), but this is probably due to the short follow-up. Taken this into account, the authors should also try to considerably mitigate their statement regarding efficacy. Furthermore, the small and specific (Japanese) population limits the generalization of the results. This should be better discussed and, probably, also added in the title. In general, I feel that, as original research article, the limitations of the work are various and of major relevance and should be better addressed.

Minor

Line 180: modify "initial assessment" with "short term assessment".

Lines 229-231: To provide a stronger meaning of safety, I suggest modifying the sentence in “In the present study, no patient presented with hypotension, even if in some patients SBP had been already normalized at baseline with other antihypertensive agents.”

Lines 233-235: The paragraph is too short. As stated above, the authors should make at least a major effort here if they intend to comment about safety. Maybe some literature quantitative contextualization would be helpful here (i.e. comparison of the effect on BNP levels - absolute or percentage reduction- with other MRB on similar cohorts). I suggest also implementing the data on hospitalization for heart failure here, and also discussing the short follow-up time.

Line 234: change “in the initial assessment” with “over time”.

Reviewer #2: The Iwahana and co-authors manuscript aimed to investigate the short- to mid-term safety and efficacy of esaxerenone in hypertensive patients with heart failure with reduced ejection fraction (HFrEF). The present study was a single-center, retrospective observational study conducted in Chiba University Hospital, included a small number of patients and part of then were converted to esaxerenone from another MRB – this observation was pointed in the discussion section. Although these points, the present study suggested that esaxerenone is safe for hypertensive patients with HFrEF, with a low incidence of hyperkalemia and hypotension. In addition, esaxerenone was found to be beneficial for heart failure in terms of BNP level reduction.

Major comments:

1. To reinforce the esaxerenone efficacy for heart failure the authors could compare the echocardiographic data before and after short- to mid-term esaxerenone treatment. Please, analyze these echocardiographic parameters.

2. The authors could include diastolic parameters in the echocardiographic analysis and evaluate the role of esaxerenone in these parameters.

Minor comments:

1. Please, add each Statistical analysis applied in the table and figure legends.

2. Please, change the supplemental figure to the regular figure 4.

6. PLOS authors have the option to publish the peer review history of their article (what does this mean?). If published, this will include your full peer review and any attached files.

Reviewer #1: No

Reviewer #2: No

---

## [Author Response · Author response to Decision Letter 0]

27 Sep 2021

Reviewer #1: Comment: Iwahana et al. conducted a retrospective study to investigate the safety and the efficacy of esaxerenone, a novel non-steroidal MRB in subjects with hypertension and heart failure with reduced ejection fraction. After a short term (35 days) and mid term (167 days) follow-up, no serious adverse event (hypotension, hyperkalemia, hospitalization for health failure) was observed. A small number (n=7) of patients showed a worsening of renal function, though superimposable with the one reported in other major trials and of mild clinical relevance. A slight, non-significant, elevation of potassium was also detected. In terms of efficacy, however, a consistent over time BNP reduction was assessed.

Albeit with the limitations implicit in a small retrospective study, the novelty of the drug under observation and the significance of the clinical context make the findings and the paper relevant. However, the authors should amend some flaws, mainly in the discussion section, to make it suitable for publication.

Major

Limitations: the main limits of this work lie in the combination of being a retrospective study with a very small sample size and no control group. Indeed, this substantially hinder the appreciation of an efficacy effect, and this should be wider discussed in the “study limitation”. Also, cardiovascular events were defined as outcome. However it seems, from the text, that no event was reported (except that one in Figure 1), but this is probably due to the short follow-up. Taken this into account, the authors should also try to considerably mitigate their statement regarding efficacy. Furthermore, the small and specific (Japanese) population limits the generalization of the results. This should be better discussed and, probably, also added in the title. In general, I feel that, as original research article, the limitations of the work are various and of major relevance and should be better addressed.

> We thank the reviewer for this insightful comment. As the reviewer indicated, this retrospective observational study has several limitations; the limited population and the shortness of the observation period. Indeed, esaxerenone is currently available only in Japan. Moreover, we were unable to follow-up the patients long-term, which seems to be one reason why the number of AEs was small. In previous studies regarding MRBs, however, K+ elevation and eGFR decrease were reported to be most conspicuous within the first month. We intended to investigate mainly the safety of the drug; thus, our results have a certain significance regarding safety. A large-scale, multinational, and randomized control study is required to resolve the potential bias. We added several sentences and a reference [23] in the Study limitations (Line 256-262).

Also, we revised the title of the manuscript to “Safety and Efficacy of Esaxerenone in Japanese Hypertensive Patients with Heart Failure with Reduced Ejection Fraction: a retrospective study”. 

Minor

Line 180: modify "initial assessment" with "short term assessment".

> We thank the reviewer for this valuable comment. We modified the sentence (Line 185), accordingly.

Lines 229-231: To provide a stronger meaning of safety, I suggest modifying the sentence in “In the present study, no patient presented with hypotension, even if in some patients SBP had been already normalized at baseline with other antihypertensive agents.”

> We thank the reviewer for this valuable comment. Given that we agree with your suggestion, we revised the sentence (Lines 242-244).

Lines 233-235: The paragraph is too short. As stated above, the authors should make at least a major effort here if they intend to comment about safety. Maybe some literature quantitative contextualization would be helpful here (i.e. comparison of the effect on BNP levels - absolute or percentage reduction- with other MRB on similar cohorts). I suggest also implementing the data on hospitalization for heart failure here, and also discussing the short follow-up time.

> We thank the reviewer for this pertinent comment. We referred to two appropriate articles. According to a review analyzing 16 phase III chronic HF trials, therapy-related changes in natriuretic peptides appeared modestly correlated with longer-term therapeutic effects on hospitalization for HF, but not with effects on all-cause mortality. Regarding MRBs, the ARTS-HF study reported that 30-40% of HF patients with diabetes mellitus and/or chronic kidney disease treated with eplerenone or finerenone presented with a >30% reduction of NT-proBNP. Esaxerenone in the present study demonstrated comparable BNP reduction to other MRBs, which suggests that esaxerenone is potentially beneficial for HFrEF. We added this content in Lines 247-253.

Reviewer #2: The Iwahana and co-authors manuscript aimed to investigate the short- to mid-term safety and efficacy of esaxerenone in hypertensive patients with heart failure with reduced ejection fraction (HFrEF). The present study was a single-center, retrospective observational study conducted in Chiba University Hospital, included a small number of patients and part of then were converted to esaxerenone from another MRB – this observation was pointed in the discussion section. Although these points, the present study suggested that esaxerenone is safe for hypertensive patients with HFrEF, with a low incidence of hyperkalemia and hypotension. In addition, esaxerenone was found to be beneficial for heart failure in terms of BNP level reduction.

Major comments:

1. To reinforce the esaxerenone efficacy for heart failure the authors could compare the echocardiographic data before and after short- to mid-term esaxerenone treatment. Please, analyze these echocardiographic parameters.

> We thank the reviewer for this important suggestion. A total of 40 patients underwent an echocardiography 160±82 days after starting esaxerenone. Comparison between pre- and post- echocardiographic parameters presented with LV reverse remodeling (LVDD: 61.8±7.4 mm to 58.9±8.7 mm, p=0.002, LVDS: 52.7±8.1 mm to 48.1±10.7 mm, p<0.001, LVEF 30.8±6.0% to 38.5±10.3%, p<0.001) and left atrial volume index reduction (LAVI: 46.5±21.1 ml/m2 to 39.1±17.0 ml/m2, p=0.02). E/A and E/e’ tended to improve, but not significantly (E/A 1.19±1.00 to 0.88±0.30, p=0.12, E/e’ 13.0±7.0 to 10.8±4.2, p=0.11). These results suggest that esaxerenone is potentially beneficial for systolic and, perhaps, diastolic function. However, we cannot conclude that these beneficial changes of echocardiography were entirely due to the efficacy of esaxerenone, due to the study design; retrospective, observational, and non-randomized, without a control group. In some patients, other cardioprotective agents were added at close timing. Therefore, we did not present with the comparison of pre- and post- echocardiographic data. A randomized and placebo control study is required to resolve this issue.

2. The authors could include diastolic parameters in the echocardiographic analysis and evaluate the role of esaxerenone in these parameters.

> We thank the reviewer for this valuable comment. We added the baseline echocardiographic data including LAVI, E/A, and E/e’ (Table 1).

Minor comments:

1. Please, add each Statistical analysis applied in the table and figure legends.

2. Please, change the supplemental figure to the regular figure 4.

> We thank the reviewer for this valuable comment. We revised the table and figure legends, and changed the supplemental figure to ‘Fig 4’.

---

## [Decision Letter · Decision Letter 1]

21 Oct 2021

Safety and Efficacy of Esaxerenone in Japanese Hypertensive Patients with Heart Failure with Reduced Ejection Fraction: a retrospective study

PONE-D-21-24007R1

Dear Dr. Iwahana,

We’re pleased to inform you that your manuscript has been judged scientifically suitable for publication and will be formally accepted for publication once it meets all outstanding technical requirements.

Kind regards,

Vincenzo Lionetti, M.D., PhD

Academic Editor

PLOS ONE

Additional Editor Comments (optional):

Reviewers' comments:

Reviewer's Responses to Questions

**Comments to the Author**

1. If the authors have adequately addressed your comments raised in a previous round of review and you feel that this manuscript is now acceptable for publication, you may indicate that here to bypass the “Comments to the Author” section, enter your conflict of interest statement in the “Confidential to Editor” section, and submit your "Accept" recommendation.

Reviewer #1: All comments have been addressed

Reviewer #2: All comments have been addressed

2. Is the manuscript technically sound, and do the data support the conclusions?

Reviewer #1: Yes

Reviewer #2: (No Response)

3. Has the statistical analysis been performed appropriately and rigorously? 

Reviewer #1: Yes

Reviewer #2: (No Response)

4. Have the authors made all data underlying the findings in their manuscript fully available?

Reviewer #1: Yes

Reviewer #2: (No Response)

5. Is the manuscript presented in an intelligible fashion and written in standard English?

Reviewer #1: Yes

Reviewer #2: (No Response)

6. Review Comments to the Author

Reviewer #1: The authors have brilliantly answered to all the issues raised; thus, I do not have any further comment. The paper is now, in my opinion, suitable for publication.

Reviewer #2: (No Response)

7. PLOS authors have the option to publish the peer review history of their article (what does this mean?). If published, this will include your full peer review and any attached files.

Reviewer #1: No

Reviewer #2: No

---

## [Editor Report · Acceptance letter]

29 Oct 2021

PONE-D-21-24007R1 

Safety and Efficacy of Esaxerenone in Japanese Hypertensive Patients with Heart Failure with Reduced Ejection Fraction: a retrospective study 

Dear Dr. Iwahana:

I'm pleased to inform you that your manuscript has been deemed suitable for publication in PLOS ONE. Congratulations! Your manuscript is now with our production department. 

Kind regards, 

on behalf of

Prof. Vincenzo Lionetti 

Academic Editor

PLOS ONE